# Diversification of Chemical Structures of Methoxylated Flavonoids and Genes Encoding Flavonoid-*O*-Methyltransferases

**DOI:** 10.3390/plants11040564

**Published:** 2022-02-21

**Authors:** Yuting Liu, Alisdair R. Fernie, Takayuki Tohge

**Affiliations:** 1Graduate School of Biological Science, Nara Institute of Science and Technology (NAIST), Ikoma 630-0192, Japan; liu.yuting.lr2@bs.naist.jp; 2Max-Planck-Institut für Molekulare Pflanzenphysiologie, 14476 Potsdam-Golm, Germany

**Keywords:** methoxylated flavonoids, *O*-methyltransferase, plant specialized metabolism

## Abstract

The *O*-methylation of specialized metabolites in plants is a unique decoration that provides structural and functional diversity of the metabolites with changes in chemical properties and intracellular localizations. The *O*-methylation of flavonoids, which is a class of plant specialized metabolites, promotes their antimicrobial activities and liposolubility. Flavonoid *O*-methyltransferases (FOMTs), which are responsible for the *O*-methylation process of the flavonoid aglycone, generally accept a broad range of substrates across flavones, flavonols and lignin precursors, with different substrate preferences. Therefore, the characterization of FOMTs with the physiology roles of methoxylated flavonoids is useful for crop improvement and metabolic engineering. In this review, we summarized the chemodiversity and physiology roles of methoxylated flavonoids, which were already reported, and we performed a cross-species comparison to illustrate an overview of diversification and conserved catalytic sites of the flavonoid *O*-methyltransferases.

## 1. Introduction

Flavonoids are a large group of phenolic compounds; this group encompasses flavonol, flavanone, flavone, isoflavone, flavanol and anthocyanin subgroups, which play important roles in the physiological response of plants to various biotic and abiotic stresses. The core structure of flavonoids, their “aglycone”, is composed of A, B and C rings, whilst a series of modification reactions, such as hydroxylation, glycosylation, prenylation and methylation, can enhance multiple physiological functions corresponding to both their structural diversity and tissue specificities. The hydroxylation reaction binds to the backbone in different numbers and positions, being the basis of subsequent *O*-methylation reactions. The *O*-methylation of flavonoid aglycones results in reduced molecular activity of a hydroxyl moiety and increased lipophilicity, and it modifies their intracellular compartmentation. Furthermore, *O*-methylation provides a branchpoint in the biosynthesis of diverse metabolic pathways, such as lignin and flavonoid phytoalexin biosyntheses, which promote antimicrobial activity [1,2]. In the case of anthocyanins, *O*-methylation stabilizes the B ring, increasing its water solubility and enhancing its redshift effect [3]. In terms of chemical properties, methoxylated flavonoids have been reported to be produced in secretion systems, such as trichomes and root hairs in plants, for example, pentamethylmyricetin and xanthohumol in tomato and hop glandular trichomes, respectively [4,5], and 3′-methoxypuerarin, 5-methoxygenistein and 2′-methoxychalcone in *Pueraria lobata*, yellow lupin and alfalfa roots, respectively [6,7,8]. However, methoxylated flavonoids also receive attention due to their pharmacological benefits. Flavonoids consumed by humans undergo intracellular metabolism, usually by *O*-methylation or glucuronidation, which reduces their ability to donate hydrogen atoms [9]. In the example of puerarin and its 3′-methoxy derivative, higher liposolubility and antioxidant activity [6], as well as advantages in the protection of cerebral ischemia–reperfusion injury, were reported [10]. Furthermore, the methoxylated isoflavones neovestitol and vestitol from Brazilian red propolis display anti-inflammatory and antimicrobial bioactivity, thereby being regarded as potentially suitable pharmacological ingredients [11]. The bioactivity of the major methoxylated flavonoids has been the focus of a couple of recent reviews [12,13].

*S*-adenosyl-methionine (SAM)-dependent methyltransferases are enzymes utilizing SAM as a universal methyl donor, which operates in both primary and secondary metabolism [14]. Based on its substrate specificity, SAM-dependent methyltransferases can be sub-classified as *O*-, *N*- or *C*- methyltransferases (OMT, NMT and CMT, respectively). Plant OMTs act as hydroxyl group acceptors of small molecules, such as flavonoids, alkaloids, phytoalexins, lignin precursors, simple phenols and phytohormones [15]. Flavonoid *O*-methyltransferases (FOMTs) of plants belong to class I and class II subgroups according to their molecular weight and cation dependency. The class I caffeoyl-CoA OMT (CCoAOMT) type group is of a lower subunit molecular weight (26–30 KDa) and displays cation-dependent activity. The enzymes of this subclass mostly participate in lignin, phytohormone and scent metabolism. Such class I CCoAOMT genes are, therefore, thought to have evolved under the pressure of terrestrial colonization [16]. The enzymes of the class II caffeic acid OMT (COMT) type group range in size between 37 and 43 KDa molecular weight and display cation-independent activity; they are mainly involved in phenylpropanoid and alkaloid biosyntheses [15]. For this reason, class II COMT genes are thought to have evolved later and in response to more diverse selective pressures [16]. In hybrid Populus (*P. deltoides* × *P. nigra*), phylogenetic analysis combined with comparative genomics of OMTs indicated that OMTs have evolved by gene duplication; COMT genes, COMT-like genes and several FOMTs are thought to have evolved different functions or tissue specificity via tandem gene duplication followed by subsequent diversification [17]. In *Brachypodium distachyon*, three duplicated genes encoding COMT are proposed as species-specific functional genes, since these COMTs do not locate in collinear genomic regions of rice and sorghum. Functional characterization verifies that they have lost the original catalytic activity toward caffeic acid [18]. In *Vanilla planifolia*, two COMT-like genes that evolved from a COMT gene have a novel function in catalyzing methyl gallate and flavone luteolin production [19]. Although, in most cases, FOMTs are cation-independent COMT-type enzymes, functional characterization of an anthocyanin OMT and an (iso) flavonol-6-OMT in grape, ice plant, sweet basil and soybean [20,21,22,23] revealed that CCoAOMT-like genes have diversified from CCoAOMT genes since they display high similarity in their catalytic mechanisms, as well as their activities all being cation dependent. 

With the exception of a minority of OMTs, which exhibit strict enzymatic specificity for a single substrate, plant OMTs generally accept a broad range of substrates across flavonoids, lignin precursors and alkaloids, albeit with different substrate preferences [14,24]. Such functional diversification of FOMTs provides both the challenge and the curiosity of characterizing their multiple in planta physiological functions. For example, recent studies have provided evidence that a variation in the modification enzymes flavonoid phenylacyltransferase and glycosyltransferase affect UV tolerance in Arabidopsis and rice, respectively [25,26], providing strong evidence that downstream modification of flavonoids affects environmental adaption. Since many FOMTs have been reported to be stress-induced and/or tissue-specific enzymes [23,27,28], elucidation of the roles and application values for the stress defense of methoxylated flavonoids is worthwhile. In this review, we summarize the present works regarding chemodiversity, the role of physiology, characterized function and structure basis of FOMTs, providing aspects for characterization of OMT enzymatic genes. We additionally discuss the current difficulties in studying the function of the individual metabolites of these classes. 

## 2. Chemical Diversity of Methoxylated Flavonoids in Land Plants

The structural diversity of known methoxylated flavonoids is determined by the combination of the position and number of methyl groups, which are largely limited by the distribution of hydroxyl groups. In principle, *O*-methylation can occur at any position; however, 7- and 4′-methoxylation are the most highly represented since their corresponding hydroxyl moieties are attached during the biosynthesis of the flavonoid aglycone (Figure 1). By contrast, according to the slight differences among flavonoid skeletons, some distinct methoxylated forms occur specifically within the aglycones. For example, before the step of cyclization by chalcone isomerase (CHS), naringenin chalcone has a specific hydroxyl moiety at the 2′ position differing from the other flavonoids; thus, it has a unique methoxylated form, namely, 4,4’-dihydroxy-2’,6’-dimethoxychalcone [29]. The backbone of flavonol has a hydroxyl moiety at the 3 position of the C ring; therefore, these compounds also have a distinct 3-methoxylated product compared to flavanone, flavone and isoflavone. Except for the common examples, 6-/8-/5-/2′-methoxylated flavonoids are relatively rarely discovered. Berim and Gang [30] and Koirala et al. [13] both compiled summaries of methoxylated flavonoids, including the commonly occurring methoxylated forms of flavonols, such as quercetin, kaempferol and luteolin, alongside their distribution. Here, we summarize the structural information to provide an overview of the distribution of methoxylated flavonols, flavones, isoflavones and anthocyanidins based on database data mining. This review also focuses on the unique methoxylated forms and plant species containing highly methoxylated flavonoids, which are potential substrates for novel functional FOMTs. 

Through a database survey of the KNApSAcK core system ([31]; http://www.knapsackfamily.com/knapsack_core/top.php, accessed date: 14 May 2021) (57,906 metabolites, 24,412 species and 137,333 metabolite–species pairs), 1598 flavonoids were found to be methoxylated flavonoid derivatives (with methoxylated aglycone of flavonols, flavones, isoflavones, flavanones, isoflavanones, flavans, isoflavans, flavanols and anthocyanidin) (Figure 2) found in plant species. 

In our survey and data mining, Fabaceae, Asteraceae, Lamiaceae and Rutaceae were the top four families in terms of the diversity of methoxylated flavonoids. As for the distribution of each group of methoxylated flavonoids, Asteraceae (25%), Fabaceae (15%), Lamiaceae (4%), Rutaceae (3%), Brassicaceae (2%) and Ericaceae (2%) occupied almost 50% of the variety of methoxylated flavonols (Figure 3). In the same manner, Asteraceae (29%), Lamiaceae (14%), Fabaceae (9%), Rutaceae (5%) and Plantaginaceae (2%) held over 58% of the variety of methoxylated flavones. For methoxylated anthocyanidin and isoflavones, the Fabaceae family possessed the most diverse compounds (Figure 3). Notably, the Fabaceae family was found to be rich in methoxylated isoflavones, with 78% among investigated families.

The degree of isoflavone methylation and its distribution were evaluated for each plant species. A total of 272 out of the 347 Fabaceae species investigated in KNApSAcK were found in the NCBI taxonomy database. The ratio of mono-, di- and tri-, as well as polymethoxylated isoflavones (more than four methoxyl groups), in 272 Fabaceae species were found in KNApSAcK. By combining the taxonomy relationship and metabolite information, the isoflavones with a mono-methoxyl group were found to be the most widely distributed, especially in some common studied species, such as *Medicago*, *Trifolium*, the *Cicer* genus and *Glycine max* (Figure 4). The 6-methoxylated isoflavone glycitein is one of the major isoflavones in legume plants, being highly accumulated in soybean mature seeds, the *M. sativa* plant and *Pueraria thunbergiana*. Although polymethoxylated flavonoids have been discovered, their biosynthesis mechanism, especially for those with more than three methoxylated sites, still remain largely unknown. The functional diversification of FOMTs in the sequential reaction and bi-function is a key point for the elucidation of the biosynthesis of polymethoxylated flavonoids.

## 3. Physiological Roles of Methoxylated Flavonoids in Planta

The flavonoid pathway is thought to be ancient and to have evolved during adaption from an aquatic to terrestrial habitat, thus being supposed to be involved in various physiology processes and complex stress defenses ranging from coloration to anti-pathogen activity and symbiosis [32]. Flower and fruit color has been focused on and investigated regarding pollination, UV defense and the ornamental industry. Variation in the pigment anthocyanin, which contributes greatly to coloration, can alter pollinator preference and provide ornamental values. The *O*-methylation of anthocyanin increases its water solubility, strengthens its color properties and shifts the color to being more red based on the methylation level [3]. Moreover, *O*-methylation produces the major pigment subgroups peonidin, petunidin and malvidin, which are responsible for a purple appearance. Malvidin- and petunidin-type anthocyanins have been found to accumulate in colored grape berries, cyclamen flowers, petunia flowers, purple tomato seedlings, *Nemophila menziesii* blue flowers and torenia blue petals [33,34,35,36,37,38], whilst peonidin-type anthocyanins have been found in peach flowers and peony flowers [39,40]. Given this, the corresponding anthocyanin 3′/3′5′ OMT genes are useful tools for artificial germplasm innovation, such as that being utilized in transgenic purple rose creation [41].

Based on the biological functions of flavonoids, the effect of *O*-methylation on the enhancement of antimicrobial activity has consistently been reported [12]. To promote physical defense against pathogens, the cell walls of grasses additionally contain tricin (3’,5’-dimethoxytricetin); this is incorporated into lignin polymers, which were discovered following the characterization of the bifunctional rice enzyme *OsAldOMT1* [42]. The methylation of flavonols is also considered to be essential for phytoalexin biosynthesis. In legume plants, methoxylated isoflavonols formononetin (4′-methoxydaidzein) and biochain A (4′-methoxygenistein) are the key precursors of phytoalexins, such as vestitol, medicarpin, pisatin, maackiain from *Lotus corniculatus*, *Medicago sativa*, *Pisum sativum* and *Trifolium pratense* [43,44,45,46]. The accumulation of formononetin and medicarpin by the elicitation of an *MsIOMT* overexpression line increased the resistance of alfalfa to the leaf pathogen *Phoma medicaginis* [47]. Glycitin (6-methoxydaidzin) and its derivatives are similarly greatly induced by the pathogens *Aspergillus oryzae* and *Rhizopus oligosporus* in soybean seedlings as opposed to daidzin, which accumulated in the negative control [23]. Furthermore, sakuranetin, a flavonoid phytoalexin, produced by the 7-*O*-methxylation of naringenin, rapidly responded both to UV irradiation and phytopathogen infection [28,48]. 

Methoxylated flavonoids have additionally been proposed to affect the interaction with symbiotic bacteria and plants. From the summary of ‘infection flavonoids’, which induce the Nod factor in the nodulation process in legume plants, 7- or 4′-methoxygenistins, glycitin, apigenin in soybean root and methoxychalcone in *M. sativa* and *Vicia sativa* are indicated as members of potential inducer factors [49]. Further evidence provided by single-cell sequencing in *M. truncatula* supports the important role of the nodule infection zone expression gene chalcone 2′-OMT and its corresponding product 4,4’ -dihydroxy-2’-methoxychalcone in the symbiosis process [50]. Moreover, plant endogenous flavonoids are reported to participate in modulating phytohormone oxidation and transportation during nodulation [51,52]. According to their chemical structure, flavonoids can have a completely different regulatory effect. For example, the 7,4′-dihydroxyflavone (DHF) induced by rhizobia inhibits IAA breakdown resulting in IAA accumulation for 14–48 h post-inoculation, whilst formononetin (4′-methoxydaidzein) accelerates IAA breakdown by stimulating or relieving inhibition of IAA oxidase activity [53]. Current studies focusing on plant–plant interactions suggest that methoxylated flavonoids possess allelopathic effects. In the case of aggressive ruderal plants, root secretions from *Dittrichia viscosa* contain apigenin, 6-methoxykaempferol, rhamnetin (7-methoxyquercetin), isorhamnetin (3′-methoxyquercetin) and dihydroxyquercetin; 7-methoxykaempferol and 6-methoxykaempferol reduce the root length and root biomass of lettuce seedlings, respectively [54]. The stability of these root-secreted methoxylated flavonoids means that they remain in the soil for a considerable period of time and can thus inhibit the growth of other species [55]. These reports suggest that methoxylated flavonoids often accumulate in secretion organs in order to subsequently fulfill their anti-pathogen and interaction signal functions. Although the function of flavonoids has been the subject of continuous attention, the impact of flavonoid *O*-methylation on symbiosis and pathogen interaction essentially comes from indirect evidence and needs to be subjected to systematic research. In addition, the effects of environmental conditions, including temperature, nutrition and water sufficiency, on flavonoid *O*-methylation remains to be explored.

## 4. Flavonoid-*O*-Methyltransferases Characterized in Plants 

### 4.1. Flavonoid 7-O-Methyltransferase

Corresponding to methoxylated flavonoid diversity, flavonoid-*O*-methyltransferases, which methylate different positions or substrates, are constantly being isolated [30]. For flavonoid A ring methylation, F7OMTs have been widely isolated to investigate their important role in phytoalexin biosynthesis and biotic stress response. F7OMTs from *Glycyrrhiza echinata*, *Hordeum vulgare*, *Mentha* × *piperita*, *Oryza sativa*, *M. truncatula*, *Populus deltoids* and *Ocimum basilicum* are involved in the catalysis of a variety of substrates, including isoflavones, luteolin, apigenin, quercetin and naringenin [22,28,56,57,58,59,60,61]. In *O. sativa* and *H. vulgare*, *OsNOMT* and *HvF7OMT* were found, following pathogen inoculation, to methylate naringenin and apigenin, respectively. In *M. truncatula,* although *MtIOMT1*, *MtIOMT2*, *MtIOMT3* and *MtIOMT7* all modify the 7-*O*- position, they have different substrate preferences with glycitein, daidzein, 6,7,4′-trihydroxyisoflavone and naringenin, respectively [59].

### 4.2. Flavonoid 3-O-Methyltransferase and Flavonoid 5-O-Methyltransferase

Given the requirement for 3-hydroxyl group existence, substrates of F3OMT are limited to flavonol aglycones. The *StF3OMT* isolated from *Serratula tinctorial* displayed a preference for quercetin, followed by kaempferol and myricetin, with no requirement for Mg^2+^ [61]. In wild and cultivated tomato glandular trichomes, *SlMOMT3* and *ShMOMT3* were characterized to catalyze the methylation of the aglycone, as well as the 7/3′/4′ methoxylated form of quercetin, kaempferol and myricetin, which participate in a series of methylation reactions leading to the highly methoxylated myricetin present in trichomes [4]. However, the *CrOMT1* isolated from *Catharanthus roseus* was found to display a border preference for phenylpropanoids, such as 5-hydroxyferulate, in addition to the 3-*O*- position methylation of flavonols [62]. With regard to F5OMT, few cases have been reported, including one genistein 5-OMT isolated from *Lupinus luteus* roots [7] and one multiple functional enzyme *CdFOMT5*, which catalyzes 3,3′,5,7 methylation of flavones in citrus fruit peels [63].

### 4.3. Flavonoid 6-O-Methyltransferase and Flavonoid 8-O-Methyltransferase

F6OMT and F8OMT are rarely isolated because of the rare existence of 6/8 methoxylated flavonoids requiring additional hydroxylation on these positions. The ice plant *PFOMT* was the first isolated F6OMT, which could methylate diverse flavonols and caffeoyl-CoA derivatives but only those with a vicinal dihydroxyl moiety. Phylogenetic analysis of protein sequences suggested that the *PFOMT* gene diverged from *CCoAOMT* genes. The subunit molecular weight, which ranges from 26 to 30 KDa (estimated as 26.6 KDa), and its Mg^2+^-dependent reaction indicated PFOMT to be a class I OMT, providing the clue that F6OMT might differ from flavonol/flavone OMTs decorating other positions, which usually belong to class II OMTs [21]. Several subsequent studies provided additional evidence to support this hypothesis. In *Plagiochasma appendiculatum* and *Glycine max*, the PFOMT-like proteins PaF6OMT and GmIOMT1 were identified as F6OMT catalyzing the production of scutellarein, baicalein and 6-OH daidzein, respectively. The characteristics of the small subunit size being estimated as 27.4 KDa and 26.75 KDa and their activity being cation dependent are consistent with belonging to PFOMT [23,64]. According to these known examples, F6OMTs are considered to phylogenetically cluster as a separate branch from CCoAOMTs, reacting with a vicinal hydroxyl group and belonging to class I OMTs, whose activity is cation dependent. However, in *O. basilicum*, two ObF6OMTs actually shared high similarity with F4′OMTs; their protein mass and non-cation reaction indicated them to be class II OMTs [22]. As for F8OMT, one flavone, namely, *MpOMT2*, was isolated to catalyze the 8-hydroxy-7-methoxyflavone in *M.* × *piperita* [58]. Subsequent research on *O. basilicum* also isolated the homologous gene of *MpOMT2* designated as *ObF8OMT-1*, both of which were cation-independent OMTs. Interestingly, the other class I OMT gene was designated as *ObPFOMT*, which displayed cation-dependent activity and required a vicinal hydroxyl group. It was additionally demonstrated to display an 8 methylation activity to flavone in *O. basilicum* [65].

### 4.4. Flavonoid 4′-O-Methyltransferase

In the B ring of flavonoids, 4′-methoxylation often corresponds to anti-pathogen activity. In Fabaceae, 4′-methoxylated isoflavones (formononetin and biochain A) are methoxylated isoflavones, and they are induced by pathogen infection. Several F4′OMTs, such as *MtIOMT5*, *HI4′OMT*, *PIOMT9* and *SOMT2,* corresponding to the production of 4′-methoxylated isoflavones, have been isolated from the legumes *M. truncatula*, *G. echinate*, *P. lobate* and *G. max* [56,59,66,67] and have been verified to play an important role in promoting phytoalexin biosynthesis [56,68]. Besides legume plants, flavonol isorhamnetin 4′-OMT *MpOMT4* from *M.* × *piperita*, flavone apigenin 4′OMT (*PaF4′OMT*) from *P. appendiculatum* and flavanone homoeriodictyol-4′OMT (*CrOMT6)* from *C. roseus* were also characterized [58,69,70]. Moreover, *ShMOMT2* was isolated from wild tomato trichome methylates from the 4′-*O*- and 7-*O*- positions of flavonol to produce trichome-specific methoxylated myricetins (3,7,3′,4′,5′-pentamethoxymyricetin) [71]. 

### 4.5. Flavonoid 3′/5′-O-Methyltransferase

The 3′/5′OMTs are widely found, with the most well known being the production of isorhamnetin by quercetin 3′-OMT. The CCoAOMT-like protein AtCCoAOMT7 of Arabidopsis displayed substrate specificity with a preference for 3′-*O*-methylation and specificities for flavone luteolin and flavonol quercetin in vitro. However, AtCCoAOMT7 had promiscuous positional selectivity for flavanones, eriodictyol and dihydroquercetin, which led to a mixture of a specific ratio of 3′-*O*-methylether and 4′-*O*-methylether [72]. The *PFOMT* from ice plant also had both 6-OMT activity and 3′-OMT activity with quercetagetin as substrate [21]. Similar to the low-molecular-weight protein PFOMT, the CCoAOMT-like protein SOMT-9 and the CCoAOMT protein SOMT-10 from soybean catalyzed the transfer of the methyl group to the 3′OH group of quercetin or luteolin [73,74,75]. Besides the CCoAOMT and CCoAOMT-like proteins, COMT-type proteins, for example, FOMTS, such as CaFOMT1, CaOMT2, MpOMT3, OsCOMT1, ShMOMT4 and PIOMT4, also function in the methylation of the 3′ position of flavones, flavonols and isoflavones, for example, quercetin, luteolin, myricetin and puerarin [6,42,58,76,77,78,79]. In Arabidopsis, *AtOMT1* is involved in multiple biosynthetic pathways, including flavonol, lignin, melatonin and NAD biosyntheses. *AtOMT1* displays enzymatic activity with 3′-*O*-methylation of quercetin [80] and also 3/5-*O*-methylation activity with caffeic aldehyde, 5-*O*-hydroxy-ferulic acid, 5-*O*-hydroxyconiferaldehyde and caffeoyl alcohol [81,82]. In addition, *AtOMT1* also displayed weak *N*-methylation activity with nicotinate [83] and in vitro *N*-acetylserotonin methylation activity [84]. 

In the case of anthocyanin, F3′OMTs and F3′,5′OMTs have been much characterized in studies of flower color engineering. By contrast to the other FOMTs, which prefer aglycone substrates, anthocyanidin OMTs prefer glycosylated substrates. In grape, F3′,5′OMT *VvAOMT* could only catalyze glycosylated anthocyanins and flavonols and not aglycones. The types of glycosides attached to the substrate also affected the relative specific activity of *VvAOMT* [20]. In peony, the F3′OMT *PsAOMT* mediated color spot formation by methylating cyanidin-3-*O*-glucoside to the darker peonidin-3-*O*-glucoside [40]. In purple tomato tissues, *SlAnOMT* could produce petunidin glucoside utilizing delphinidin 3-*O*-glucoside. Silencing *SlAnOMT* in fruits and hypocotyls resulted in a reduction in the content of petunidin and malvidin [36]. In research focused on soybean seed coat pigmentation, *GmOMT5* was characterized as a pigment isogene methylating cyanidin glucoside to peonidin glucoside in black seed coated compared to brown seed coated soybean seeds [85]. Besides the 3′ catalytic activity, some AOMTs, such as *NmAMT3*, *NmAMT6* and *MT2 lotus* from *Nemophila menziesii* and *Petunia x hybrida*, also display slight 3′,5′ catalytic activity as well as 3′-OMT function [37].

### 4.6. Chalcone 2′-O-Methyltransferase

Given that chalcones are missing a C ring in their chemical structure, chalcone OMTs are a unique class methylated at the 2′-*O*-position. The isoliquiritigenin 2′-OMT from alfalfa has been postulated to play a role in the nodulation process due to it producing the *Nod* gene inducer 4,4’-dihydroxy-2’-methoxychalcone in the rhizobia infection area of root hairs [51,86]. Furthermore, in *Humulus lupulus*, which is used as an ingredient for beer brewing, a chalcone 6′-OMT (equal to 2′ position) *OMT1* was characterized to produce the flavor compound xanthohumol from desmethylxanthohumol in trichomes [5]. 

### 4.7. Flavonoid O-Methyltransferase with Multiple-Site Selectivity

In addition to the FOMTs mentioned above (4.1 to 4.6), some FOMTs show methylation activity for multiple sites, especially in plant species that produce a high level of methoxylated flavonoids, such as citrus and wheat. The peel of citrus fruits contains high contents of methoxylated flavones, with up to seven methoxylation sites (3,5,6,7,8,3′,4′-heptamethoxyflavone) [87]. *CdOMT5* and *CrOMT2* from *C. depressa* and *C. reticulata* were characterized to methylate at the 3,3′,5,7 and 3′,5′/7 positions with a wide range of flavones or flavonols, respectively [64,88]. In wheat, *TaOMT1* and *TaOMT2* were found to sequentially methylate tricetin via 3′-*O*-monomethyl ether (selgin) to 3′,5′-dimethyl ether (tricin) to 3′,4′,5′-trimethyl ether order [89,90]. 

A set of FOMTs with different regioselectivity and substrate preferences have been characterized in same species, such as isoflavone OMTs *MtIOMT1-7* from *M. truncatula*; flavonol OMTs *ShMOMT1-4* from *S. habrochaites*; flavone OMTs *MpOMT1A, 1B* to *MpOMT4* from *M.* × *piperita;* and flavone OMTs *ObFOMT1-6*, *ObF8OMT-1* from *O. basilicum* [4,22,58,59,79], providing proof that FOMTs show different functions with shifted substrate selectivity in spite of sharing high sequence similarity. By phylogenetic analysis of characterized FOMTs, the regioselectivity of candidate FOMTs can be predicted, but only for genes classified in the clade of functionally characterized OMTs. In addition, FOMTs may have different substrate selectivity in vivo compared to in vitro because of the complex internal environment, such as substrate spatiotemporal existence or the rapid glycosylation of compounds. Given this, the determination of enzyme activity by in vitro experiments is required. In *M. truncatula* elicited leaves, over-expressing *MsI7OMT* did not produce isoformononetin (7-methoxydaidzein) but rather accumulated formononetin (4′-methoxydaidzein) [47]. In camptotheca, an alkaloid biosynthetic enzyme, 10-hydroxycamptothecin *O*-methyltransferase, could also methylate the 7-*O*-position of kaempferol and quercetin aglycone in vitro; however, no 7-methoxylated flavonoids were detected in vivo, indicating that the only in vivo substrate of this enzyme was 10-hydroxycamptothecin [24].

## 5. The Structural Basis of Flavonoid *O*-Methyltransferase Function 

The FOMTs belonging to class I and class II OMTs harbor conserved catalytic sites and substrate binding regions but are separately discussed according to their differences in length, conserved amino acid sites and cation interaction (Figure 5). In 1998, Joshi and Chiang [91] conducted a study to improve the mismatch of the conserved region in silico by utilizing more than 10 subgroups of OMT sequences from 56 plant species. As a result, consensus sequence motif A (V/I/L)(V/L)(D/K)(V/I)GGXX(G/A), motif B (V/I/F)(A/P/E)X(A/P/G)DAXXXK(W/Y/F) and motif C (A/P/G/S)(L/I/V) (A/P/G/S)XX(A/P/G/S)(K/R)(V/I)(E/I)(L/I/V) were indicated as potential SAM binding sites across class I and class II OMTs; however, to date, there is a lack of experimental support for these findings. In a later study focusing on the class I caffeoyl-CoA OMT of alfalfa, alignment using available crystal structures showed a SAM-binding site, a caffeoyl-CoA-binding site and a dimerization region [92]. Further research on anthocyanin OMTs compared the sequence to caffeoyl-CoA OMTs, reaching the conclusion that the substrate binding sites 21Lys(K)-61Met(M)-163Asp(D)-190Asn(N)-193Trp(W)-206Arg(R)-208Tyr(Y)-212Tyr(Y); SAM binding sites 58Trp(W)-85Glu(E)-87Gly(G)-93Ser(S)-111Asp(D)-140Ala(A)-165Asp(D); and dimerization region residues 27-42 (D/EALXXYI/LL/FETSV/AY/FPRE), residues 66-78 (DEG/AQ/LFL/IS/NM/LL/FLKL/IX) and residues 216-225 (V/LL/ME/KXNK/SA/FLAX) were almost conserved in caffeoyl-CoA OMTs, whereas residues 50-64 and 200-215 containing the substrate recognition sites were not conserved [35]. Additionally, a catalytic triad, namely, 166Lys(K)-190Asn(N)-238Asp(D), was newly characterized, and it was verified by mutagenesis experiments to be essential for efficient catalytic capacity of class I OMTs [93] (Figure 6A), as well as being conserved in isoflavone 6-OMT *GmI6OMT* from soybean [23]. 

Of the class II OMTs, the chalcone OMT and isoflavone OMT from alfalfa were the first two reported crystal structures of plant OMTs, revealing their substrate specificity (Figure 6). The structure basis and sequence alignment of known class II OMTs indicate 194Asp(D)-196Gly(G)-219Asp/Glu(D/E)-220Arg/Leu/Gln(R/L/Q)-239Asp(D)-240Met(M)-253Lys(K)-259Trp(W) residues to be SAM binding sites, and 257His(H)-288Asp/Glu(D/E)-318Glu/Val(E/V) residues to be involved in catalysis. Further mutation of 257His causes a failure to generate a corresponding product. Unlike the relatively conserved catalytic sites, substrate binding residues at 117, 307, 310 and 314 positions present diversity in accordance with substrate discrimination, except 168Met(M) and 311Met(M), which are thought to help constrain the A ring [94]. Based on the findings of Zubieta, studies on sweet basil and wheat support the conservation of catalytic residues in *ObaCVOMT1, ObaEOMT1* and *TaCOMT-3D* [95,96].

**Figure 5 plants-11-00564-f005:**
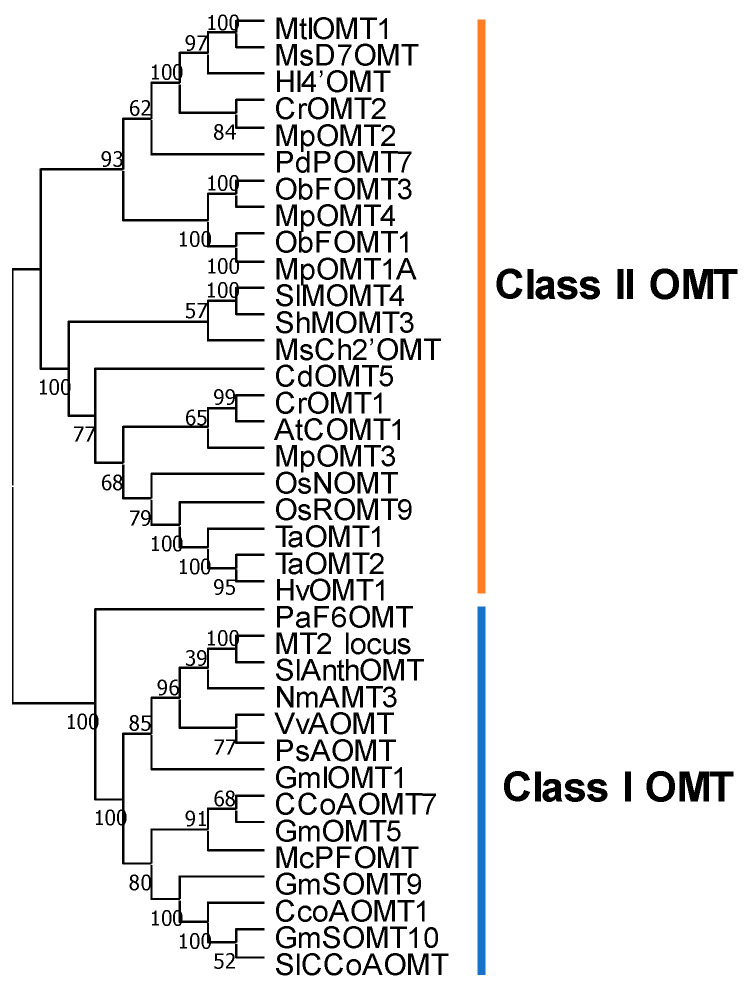
Phylogenetic tree of representative OMT genes. Class I OMT genes, including caffeoyl CoA OMTs and anthocyanin OMTs; class II OMTs, including caffeic acid OMTs, flavonoid OMTs and isoflavone OMTs. The tree was built by MEGAX [97]. The neighbor-joining method was used for clustering. The percentages of replicate trees in the bootstrap test (1000 replicates) are shown next to the branches. The evolutionary distances were computed using the p-distance method and are displayed in the units of the number of amino acid differences per site. Accession numbers: CCoAOMT1, At4g34050; CCoAOMT7, At4g26220; GmSOMT9, Glyma.17g171100; GmSOMT10, Glyma.07G214700; GmIOMT1, Glyma.05g14700; GmOMT5, Glyma.05G223400; VvAOMT. NP_001290011; McPFOMT, AY145521; PsAOMT, no ID; NmAMT3, LC330945; MT2 lotus, KJ676515.1; SlAnthoOMT, Solyc09g082660; SlCCoAOMT, Solyc02g093270.2; PaF6OMT, no ID; AtCOMT1, At5g54160; MsCh2′OMT, L10211; MpOMT1A, AY337457; MpOMT2, AY337459; MpOMT3, AY337460; MpOMT4, AY337461; MsD7OMT, U97125; HI4′OMT, AB091684; HvOMT1, ABQ58825; CrOMT1. AY028439; CrOMT2, AAM97497; TaOMT1, Q84N28; TaOMT2, ABB03907; OsNOMT, BAM13734; OsROMT9, ABB90678; ShMOMT3, AGK26768; SlMOMT4, KF740343; MtIOMT1, AY942159.1; CdOMT5, LC126059; ObFOMT1, K0I977; ObFOMT3, K0I7Q2; PdPOMT7, TC29789.

**Figure 6 plants-11-00564-f006:**
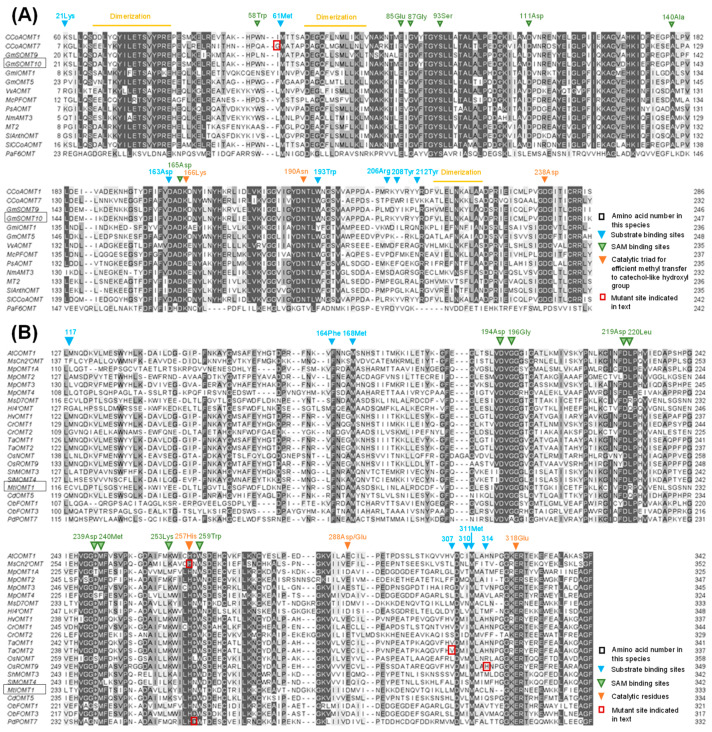
Multiple alignment of representative OMT genes. (**A**) Multiple alignment of class I OMT genes, including caffeoyl-CoA OMTs and anthocyanin OMTs. (**B**) Multiple alignment of class II OMTs, including caffeic acid OMTs, flavonoid OMTs and isoflavone OMTs. Blue inverted triangle indicates substrate binding sites, green inverted triangle indicates SAM binding sites, orange inverted triangle indicates catalytic residues, and red box indicates mutant site mentioned in this paper. Accession numbers shown in Figure 5.

Along with the increasing number of elucidated crystal structures of FOMTs, homology modeling optimized by known FOMTs with docked substrates provided a prediction of the possible residues that affect activity and selectivity. The Gly46 in AtCCoAOMT7, which is equivalent to Tyr51 in known PFOMTs, was indicated via docking studies to be a part of the substrate binding pocket close to the catalytic site. The mutation of Gly46 to Tyr led to a reverse in the *para*- to *meta*-*O*-methylation of flavanones and dihydroflavonols in vitro [72] (Figure 6A). The substitution of 328-His with Arg in the ROMT9 gene product changed the hydrophobic pocket, resulting in a regioselectivity shift from 3′,5′ to 3′ hydroxyl groups [98]. Moreover, a single amino acid mutation, Asp257Gly, in the flavonol 7-*O*-methyltransferase POMT7 protein allowed the methylation of both 3, 7-hydroxyl groups of quercetin instead of the 3-hydroxyl group alone [99]. The Val309 of TaOMT2 in wheat, which is next to the catalytic site His262 in homology modeling, decides the substrate preference for tricetin (Figure 6B). The Val309Ile TaOMT2 mutant in wheat and Ile316Val MtCOMT mutant in *M. trunculata* alter the substrate preference from a higher tricetin and 5HFA (5-hydroxyferulic acid) affinity to a higher 5HFA and tricetin affinity, respectively [100]. Moreover, studies on sweet basil and *M. truncatula* characterized a set of FOMTs sharing high similarity; however, having a different substrate preference and regioselectivity can also provide clues for essential residue identification [22,59]. Besides the variance in substrate binding sites, the cations have also been proved to dramatically modulate the substrate specificity of class I OMTs [101]. 

The protein crystal structures of PaMTH1, MsChOMT, MsIOMT and HI4′OMT suggest that the homodimer formation generated by the N-terminal swapping of FOMT protein forms the functional homodimers in solution or the cell [94,102,103]. A report on flavone *O*-methyltransferase in wheat compared the homodimer and dissociated monomers with the dissociated monomer and concluded that the monomers retain their catalytic capacity [104]. Nevertheless, several studies referring to alkaloid biosynthesis suggested that heterodimers may contribute to the catalysis of new substrate. In keeping with this, complex heterodimers of four OMT proteins from *Thalictrum tuberosum* showed selectivity to a variety of new substrates from catechols to hydroxycinnamates and alkaloids compared to its corresponding homodimers [105]. A recently discovered heterodimer consisting of PsSOMT2 and PsSOMT3 or Ps6OMT filled the missing step for noscapine biosynthesis in *Papaver somniferum* [106]. According to such evidence, elucidating the function of the heterodimers of FOMT in planta may lead to the elucidation of unknown biochemical processes.

## 6. Concluding Remarks and Future Prospects

Flavonoids are phytochemicals involved in pathogen defense and UV light protection, and they enable plants to interact with their environment. Evolutionary analysis of flavonoid biosynthesis suggests that this pathway originated very early in plant colonization of land. However, the lignin biosynthesis pathway, which is also derived from phenylpropanoid metabolism, as well as flavonoid biosynthesis, is considered to be an essential factor of land colonization of plants. Given that many FOMTs have multiple preferences toward both flavonoid and lignin biosyntheses metabolites, the diversification and convergence of substrate selectivity and physiological functions involving multiple pathways are a topic for future research. Due to the multiple substrate selectivities caused by protein dimerization and the difference in activity performance in planta and in vitro, a comprehensive elucidation covering all substrate specificities and their interaction is very complex. Therefore, a comprehensive approach combining profiling data of endogenous methoxylated/non-methoxylated compounds and a gene expression analysis considering their tissue specificity and stress deducibility of FOMTs is required in future. In this review, we summarize our current knowledge of the chemical diversity and physiological roles of methoxylated flavonoids. Additionally, we provide a cross-species comparison of methoxylated flavonoids and FOMT genes. Such plant-species-wide approaches will give an overview of the diversification of the *O*-methylation of specialized metabolism in the plant kingdom. 

## Figures and Tables

**Figure 1 plants-11-00564-f001:**
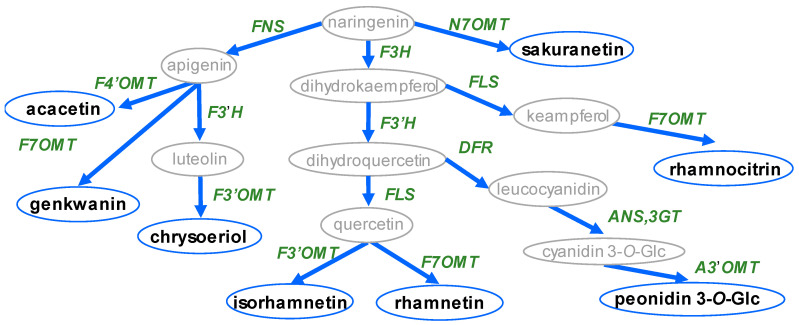
Schematic diagram of known methoxylated flavonoid aglycone biosynthesis. Circles indicate metabolite name, arrows indicate biosynthesis pathway, and words in *italics* indicate biosynthesis genes. Abbreviations: FNS: flavone synthesis; F3H: flavanone 3-hydroxylase; FLS: flavonol synthase; F3′H: flavonoid 3′-hydroxylase; DFR: dihydroflavonol 4-reductase; ANS: anthocyanin synthase; 3GT: flavonoid 3-*O*-glucosyltransferase; F7OMT: flavonoid 7-*O*-methyltransferase; F3′OMT: flavonoid 3′-*O*-methyltransferase; F4′OMT: flavonoid 4′-*O*-methyltransferase; A3′OMT: anthocyanin 3′-*O*-methyltransferase; N7OMT: naringenin 7-*O*-methyltransferase.

**Figure 2 plants-11-00564-f002:**
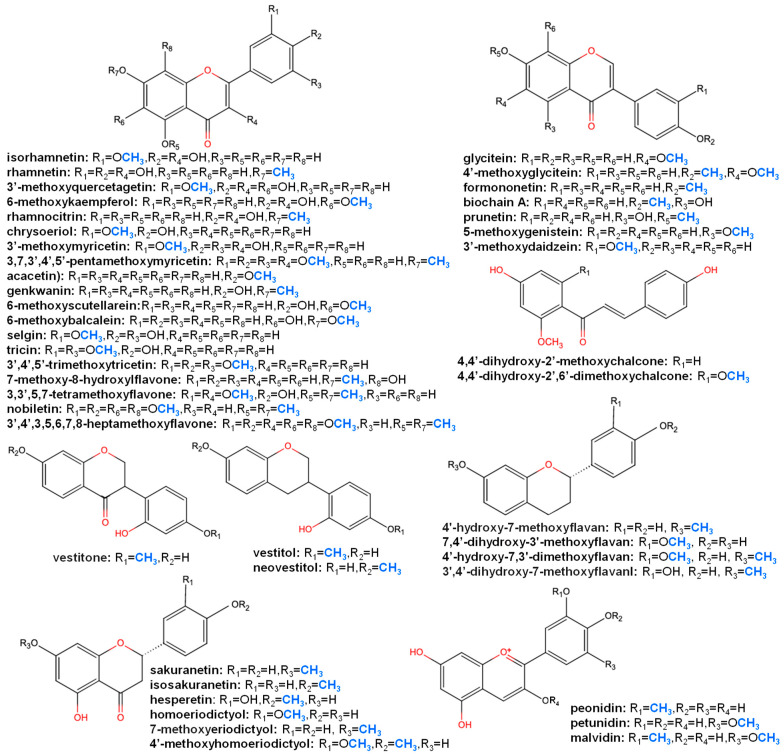
Chemical structure of major methoxylated flavonoid aglycones.

**Figure 3 plants-11-00564-f003:**
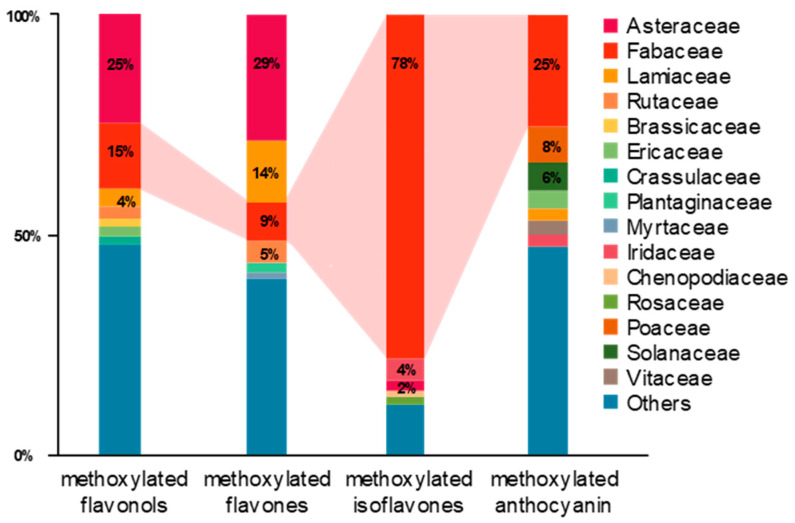
Methoxylated flavonoids reported in plant species. By searching the keywords “quercetin”, ”kaempferol”, “isorhamnetin”, “myricetin”, “flavonol”, “luteolin”, “tricetin”, “apigenin”, “flavone”, “daidzein”, “glycitein”, “genistein”, “isoflavone”, “petunidin”, “malvidin”, “peonidin” and “anthocyanidin” in the KNApSAcK database (http://www.knapsackfamily.com/knapsack_core/top.php, accessed date: 14 May 2021), 322 out of 1046 flavonols, 788 out of 1298 flavones, 349 out of 573 isoflavones and 120 out of 506 anthocyanidins were found to be methoxylated flavonoids.

**Figure 4 plants-11-00564-f004:**
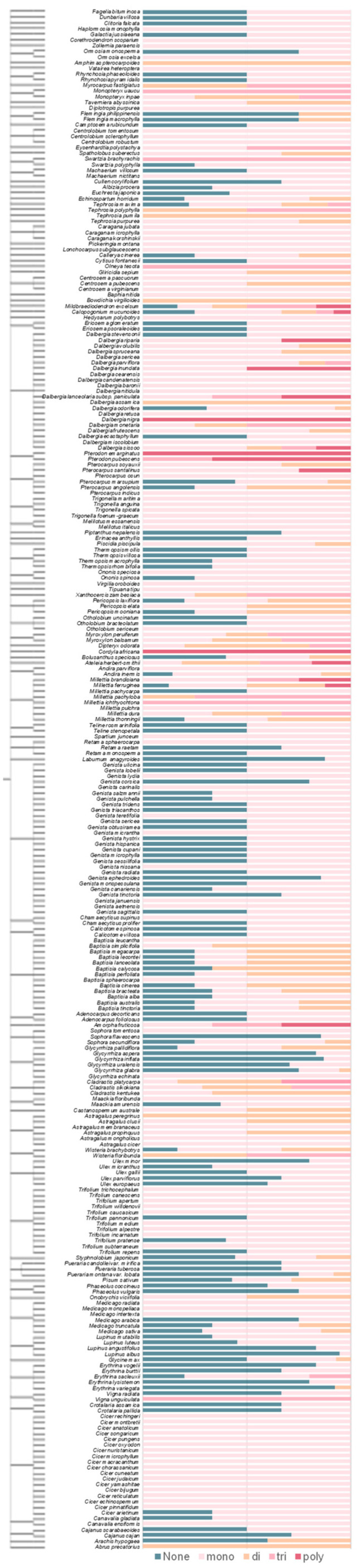
Chemical diversity and species specificity of methoxylated flavonoids in Fabaceae plants.

## Data Availability

Not applicable.

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
