# Peer review of "Diversification of Chemical Structures of Methoxylated Flavonoids and Genes Encoding Flavonoid-O-Methyltransferases"

_plants, 2022, doi:10.3390/plants11040564_

Round 1
Reviewer 1 Report
The manuscript entitled "Diversification of chemical structures of methoxylated-flavonoids and genes encoding flavonoid-O-methyltransferases" covers an interesting area of methylated flavonoids plants.
It is weel structured and written mansucript. However, some sections need revision as suggested below:
- I suggest authors to provide an schematic diagram of synthesis of flavonoids fro the explanation provided in Line 105-111 taking one of the methoxylated flavonoid as example.
- Figure 1 and Line 122-125, authors mentions that they found only 35 methoxylated flavonoids. Is this the limitation of KNApSAcK core system? Authors have also mentioned hundreds of other methoxylated flavonoids in following sentences from Line 125. Ifthat is limitation, that should be clearly mentioned.
- Line 125, In our survey....what search engine did authors use?
- There are many other methoxylated flavonoids such as citrus polymethoxyflavonoids eg. nobiletin, hesperetin and many which are important ones.
- Similarly, tricetin is not a methoxylated flavonoid.
- The name 3,7,3’,4’,5’-methyl-myricetin is not correct. Should not it be 3,7,3’,4’,5’-pentamethyl-myricetin. Same with 3,3’,5,7-methyl-flavone and 3’,4’,3,5,6,7,8-methyl-flavone.
- Similarly, for 4,4’-dihydroxy-2’,6’-methoxychalcone why it is dihydroxy but not dimethoxy?
- Authors have missed methoxylated derivatives of flavanols such as catechin and flavans.
- For peonidin and other two anthocyanidins, authors have provided names of anthocyanidins but structure contains Glc in 3-position, thus are anthocyanins. Please correct teh structure. Please check all other structures in detail once agian and correct if necessary.
- First four compounds in Fig.1 have OCH3 in blue font and other do not have. Make all in same pattern.
- Fig.3, If I a not mistaken, I think all these plant species belong to Fabaceae. If so, it is better to mention that in Figure caption as Chemical diversity and species specificity of methoxylated-flavonoids in Fabaceae plants.
- Please make all these scientifi names in Fig.3 in italics.
- It looks liek manuscript was submitted without proper final check. There are few comments like (add references) in Line 222 and cite apper also in 122. Please chqack carefully before submission.
- Do environemntal conditions such as temperature, sunlight, plant nutrition, pest infestations and diseases have role in flavonoid methoxylation? Brief explanation is necessary to understand their physiological role in detail.
- This study mainly focuses on methoxylated aglycones. How about the glycosylated ones? For example, does glycosylation occure first or methoxylation in general?
Author Response
It is well structured and written manuscript. However, some sections need revision as suggested below:
- I suggest authors to provide an schematic diagram of synthesis of flavonoids for the explanation provided in Line 105-111 taking one of the methoxylated flavonoid as example.
>> Thank you very much. A schematic diagram of biosynthesis is shown in Figure 1, as suggested.
- Figure 1 and Line 122-125, authors mentions that they found only 35 methoxylated flavonoids. Is this the limitation of KNApSAcK core system? Authors have also mentioned hundreds of other methoxylated flavonoids in following sentences from Line 125. If that is limitation, that should be clearly mentioned.
>> The “35” was revised as “1579”, thank you for pointing out. Figure 1 shows major methoxylated-flavonoid aglycones,to avoid misunderstandings, we have modified figure 1 and figure legend. We have also carefully checked the number of compounds presented in this manuscript.
- Line 125, In our survey....what search engine did authors use?
>> In this phytochemical survey, we used KNApSAcK core system (http://www.knapsackfamily.com/knapsack_core/top.php) for data mining. We used google chrome as my search engine, but any search engines are OK for this survey. Hopefully this answer will reach the reviewer’s point.
- There are many other methoxylated flavonoids such as citrus polymethoxyflavonoids eg. nobiletin, hesperetin and many which are important ones.
>> Revised. Thank you very much for your comment. These methoxylated flavonoids are accidentally missing in this Figure 1.
- Similarly, tricetin is not a methoxylated flavonoid.
>> We revised the “tricetin” in figure 1 as “3’,4’,5’-trimethoxytricetin”.
- The name 3,7,3’,4’,5’-methyl-myricetin is not correct. Should not it be 3,7,3’,4’,5’-pentamethyl-myricetin. Same with 3,3’,5,7-methyl-flavone and 3’,4’,3,5,6,7,8-methyl-flavone.
>> Thank you, we unified the name as “position-xxxmethoxyflavonoid” in the text and figure.
- Similarly, for 4,4’-dihydroxy-2’,6’-methoxychalcone why it is dihydroxy but not dimethoxy?
>> We unified the name in text and figure.
- Authors have missed methoxylated derivatives of flavanols such as catechin and flavans.
>> We did not include methoxylated flavan-ols. Now we added them to the count of methoxylated flavonoids. We also revised text. Thank you very much.
- For peonidin and other two anthocyanidins, authors have provided names of anthocyanidins but structure contains Glc in 3-position, thus are anthocyanins. Please correct teh structure. Please check all other structures in detail once again and correct if necessary.
>> Thank you, we revised the “peonidin” as “peonidin 3-O-Glc", the same to “petunidin” and “malvidin” in figure 1.
- First four compounds in Fig.1 have OCH3 in blue font and other do not have. Make all in same pattern.
>> We unified the format as you suggested.
- 3, If I a not mistaken, I think all these plant species belong to Fabaceae. If so, it is better to mention that in Figure caption as Chemical diversity and species specificity of methoxylated-flavonoids in Fabaceae plants.
>> We revised the figure legend as you suggested.
- Please make all these scientifi names in Fig.3 in italics.
>> We revised the scientific names in fig.3 in italics
- It looks liek manuscript was submitted without proper final check. There are few comments like (add references) in Line 222 and cite apper also in 122. Please check carefully before submission.
>> Thank you, we checked and revised in the main text.
- Do environmental conditions such as temperature, sunlight, plant nutrition, pest infestations and diseases have role in flavonoid methoxylation? Brief explanation is necessary to understand their physiological role in detail.
>> There some reports show that FOMT can response to biotic stress like microbial or pest stress. The anthocyanin OMT will affect the flower color, so the flower color pattern can also affect the pollination pattern recognized by pollinator. As for plant nutrition/sunlight, not so much evidence supports the relationship between flavonoid methoxylation and nutrition/sunlight. As for temperature, no direct evidence was collected, but glycitin seems more stable than daidzin and genistein at elevated temperature according to data, however not so obvious. Thank you for your comment, we added these points in line 207-208.
- This study mainly focuses on methoxylated aglycones. How about the glycosylated ones? For example, does glycosylation occur first or methoxylation in general?
>> In general, the FOMTs catalyze flavonoid aglycone instead of glycosylated flavonoids in vitro experiment, however, anthocyanin OMTs are special, because they usually have higher preference for catalyzing glycosylated anthocyanin instead of anthocyanidin in vitro experiment. We revised text line 285-293.
Reviewer 2 Report
The review is well organized and comprehensive. I only have one comment that the authors should add some phynogenetic trees to show the differences among different methyltransferase, and also discuss the different cluters with different functions.
Author Response
The review is well organized and comprehensive. I only have one comment that the authors should add some phynogenetic trees to show the differences among different methyltransferase, and also discuss the different cluters with different functions.
> Thank you very much for your reviewing and comment. Based on a suggestion from a reviewer, we have added a new phylogenetic tree of known FOMTs in Figure 4.
Round 2
Reviewer 1 Report
Authors have revied and improved the manuscript.
It can be accepted in present form.